# Lowered Quality of Life in Long COVID Is Predicted by Affective Symptoms, Chronic Fatigue Syndrome, Inflammation and Neuroimmunotoxic Pathways

**DOI:** 10.3390/ijerph191610362

**Published:** 2022-08-19

**Authors:** Michael Maes, Haneen Tahseen Al-Rubaye, Abbas F. Almulla, Dhurgham Shihab Al-Hadrawi, Kristina Stoyanova, Marta Kubera, Hussein Kadhem Al-Hakeim

**Affiliations:** 1Department of Psychiatry, Faculty of Medicine, Chulalongkorn University, Bangkok 10330, Thailand; 2Department of Psychiatry, Medical University of Plovdiv, 4002 Plovdiv, Bulgaria; 3Research Institute, Medical University Plovdiv, 4002 Plovdiv, Bulgaria; 4School of Medicine, Barwon Health, IMPACT, The Institute for Mental and Physical Health and Clinical Translation, Deakin University, Geelong 3217, Australia; 5College of Medical laboratory Techniques, Imam Ja’afar Al-Sadiq University, Najaf 54001, Iraq; 6Medical Laboratory Technology Department, College of Medical Technology, The Islamic University, Najaf 54001, Iraq; 7Al-Najaf Center for Cardiac Surgery and Transcatheter Therapy, Najaf 54001, Iraq; 8Laboratory of Immunoendocrinology, Department of Experimental Neuroendocrinology, Maj Institute of Pharmacology, Polish Academy of Sciences, 12 Smetna St., 31-343 Krakow, Poland; 9Department of Chemistry, College of Science, University of Kufa, Kufa 54002, Iraq

**Keywords:** Long COVID, depression, myalgic encephalomyelitis/chronic fatigue syndrome, depression, neuro-immune, inflammation, psychiatry

## Abstract

The physio-affective phenome of Long COVID-19 is predicted by (a) immune-inflammatory biomarkers of the acute infectious phase, including peak body temperature (PBT) and oxygen saturation (SpO2), and (b) the subsequent activation of immune and oxidative stress pathways during Long COVID. The purpose of this study was to delineate the effects of PBT and SpO2 during acute infection, as well as the increased neurotoxicity on the physical, psychological, social and environmental domains of health-related quality of life (HR-QoL) in people with Long COVID. We recruited 86 participants with Long COVID and 39 normal controls, assessed the WHO-QoL-BREF (World Health Organization Quality of Life Instrument-Abridged Version, Geneva, Switzerland) and the physio-affective phenome of Long COVID (comprising depression, anxiety and fibromyalgia-fatigue rating scales) and measured PBT and SpO2 during acute infection, and neurotoxicity (NT, comprising serum interleukin (IL)-1β, IL-18 and caspase-1, advanced oxidation protein products and myeloperoxidase, calcium and insulin resistance) in Long COVID. We found that 70.3% of the variance in HR-QoL was explained by the regression on the physio-affective phenome, lowered calcium and increased NT, whilst 61.5% of the variance in the physio-affective phenome was explained by calcium, NT, increased PBT, lowered SpO2, female sex and vaccination with AstraZeneca and Pfizer. The effects of PBT and SpO2 on lowered HR-QoL were mediated by increased NT and lowered calcium yielding increased severity of the physio-affective phenome which largely affects HR-QoL. In conclusion, lowered HR-Qol in Long COVID is largely predicted by the severity of neuro-immune and neuro-oxidative pathways during acute and Long COVID.

## 1. Introduction

The coronavirus disease 2019 (COVID-19) caused by the severe acute respiratory syndrome coronavirus 2 (SARS-CoV-2) is still spreading around the world. Most people with COVID-19 have mild clinical symptoms, but some people may experience acute respiratory distress or even severe acute respiratory syndrome (SARS), which can lead to multiorgan failure and death, especially in older adults and people with comorbid conditions such as hypertension, obesity, and type 2 diabetes mellitus (T2DM) [1,2]. Ten to twenty percent of COVID-19 patients will have Long COVID symptoms within weeks to months of recovery, whether they experienced symptoms or were asymptomatic during the acute stage of the illness [3,4]. Following recovery from the acute phase, many people with Long COVID experience a range of mental symptoms, such as sleep disturbances, affective symptoms (low mood and anxiety), chronic fatigue, neurocognitive impairments, as well as somatic manifestations such as muscle pain, muscle tension, autonomic symptoms, gastro-intestinal symptoms, headache, and a flu-like malaise [5,6,7,8,9,10,11].

In recent studies, we discovered that (a) acute and Long COVID are both characterized by concurrent elevations in fatigue and physiosomatic (previously called psychosomatic) symptoms (pain symptoms, malaise, muscle tension, somatic anxiety, gastrointestinal anxiety, genitourinary anxiety, somatic sensory, cardiovascular, autonomous and genitourinary symptoms), key depression (depressed mood, guilt-related feelings, suicidal ideation, loss of interest), and key anxiety (anxious mood, tension, fears, anxiety behavior at interview) symptoms; and (b) that in both conditions a validated and replicable single latent vector could be extracted from these depression, anxiety, fatigue and physiosomatic symptoms [7,8,12,13]. This indicates that a single latent trait drives these multiple neuro-psychiatric symptoms which, therefore, are all manifestations of the common core dubbed the “physio-affective phenome” of both acute and Long COVID [7,8,12,13].

The pathogenesis of acute COVID-19 involves the entry of SARS-CoV-2 into host respiratory epithelial cells followed by viral replication and translation in the cytoplasm and infection of adjacent cells of the host cells [14,15,16,17,18,19,20,21]. These processes are accompanied by the activation of immune-inflammatory pathways and, at times, lung injuries, pneumonia, and excessive inflammatory responses, which can progress into disseminated intravascular coagulation and multisystem failure [14,15,16,17,18,19,20,21].

The physio-affective phenome in acute and long COVID-19 is substantially predicted by immune-inflammatory pathways. First, abnormalities from chest computerized tomography scans (CCTAs), lower oxygen saturation in peripheral blood (SpO2), increased peak body temperature (PBT), and higher levels of immune-proinflammatory mediators are all strongly associated with the physio-affective phenome in acute COVID [12,13]. Importantly, during acute infection, lower SpO2 and higher PBT are reliable indicators of the intensity of the immune-inflammatory response, and they may even be able to predict the development of critical COVID-19 and higher mortality [13,22]. Second, indices of increased nitro-oxidative and immune-inflammatory processes, such as C-reactive protein (CRP), myeloperoxidase (MPO), nitric oxide, lipid and protein oxidation, activation of the nucleotide-binding domain, leucine-rich repeat and pyrin domain-containing protein 3 (NLRP3) inflammasome, and lowered antioxidant defenses, such as lowered glutathione peroxidase (Gpx) and zinc levels, and lowered serum calcium are also strongly predictive of the physio-affective phenome of Long COVID [7,8]. The physio-affective phenome of long COVID is, therefore, substantially predicted by the infectious-immune-inflammatory pathways during acute SARS-CoV-2 infection, and these effects are mediated via immune-inflammatory and nitro-oxidative pathways, according to our findings [7,8]. It should be stressed that also in major depression (MDD), bipolar disorder (BD), and chronic fatigue syndrome/myalgic encephalomyelitis (CFS/ME), the same pathways and biomarkers are substantially associated with the severity of these disorders [23,24,25].

Lowered health-related quality of life (HR-QoL) is another characteristic of Long COVID. For instance, 90 days after the acute phase, patients exhibiting symptoms of Long COVID (such as fatigue, sadness, myalgia, joint pain, dyspnea, anxiety, and vertigo) had significantly poorer SF-36 ratings for physical function, vitality, mental health, and physical health [26]. Another study found that HR-QoL was substantially lower 8.1 (±3.2) months after the acute episode [27]. In both systematic review and meta-analysis studies, HR-QoL was considerably decreased in Long COVID-19 patients especially in those with fatigue and admission to critical care units, with up to 37% (95% confidence interval: 18 to 60%) of the patients exhibiting decreased HR-QoL [28,29,30]. In neuropsychiatric disorders, including affective disorders and schizophrenia, lowered HR-QoL is substantially linked with the severity of depression, chronic fatigue, anxiety, and physiosomatic symptoms, as well as immune-inflammatory and nitro-oxidative pathways [31,32,33]. However, there are no data indicating whether the immune-inflammatory response during the acute phase and/or the subsequent immunological and nitro-oxidative pathways of Long COVID may explain the decreased HR-QoL.

Hence, the present study was performed to delineate the effects of the inflammatory responses during the acute phase (assessed using PBT and SpO2), and indicants of increased neurotoxicity during Long COVID (including activation of the NLRP3 and oxidative pathways, insulin resistance and lowered serum calcium) on four different domains of HR-QoL (physical, psychological, social and environmental) and whether the effects of the acute infectious phase on the Long COVID physio-affective phenome are mediated by increased neurotoxicity, insulin resistance and lowered calcium.

## 2. Methods

### 2.1. Participants

Prior to participating in our study, we obtained written informed consent from all participants. The institutional ethics board and the Najaf Health Directorate-Training and Human Development Center approved our research, which were numbered 8241/2021 and 18378/2021, respectively. The current study was designed and carried out in accordance with Iraqi and international ethical and privacy laws, including the World Medical Association’s Declaration of Helsinki, The Belmont Report, the Council for International Organizations of Medical Sciences (CIOMS) Guideline, and the International Conference on Harmonization of Good Clinical Practice; our institutional review board follows the International Guidelines for Human Research Safety; and our institutional review board adheres to the International Guidelines for Human Research Safety (ICH-GCP).

In this study, we recruited 86 participants with Long COVID and 39 normal controls from September to the end of December 2021. Long COVID was diagnosed using the World Health Organization (WHO) criteria [3]. These criteria used here are as follows: (a) the subjects should have a confirmed infection with COVID-19, (b) patients’ daily life activities should be influenced by at least two symptoms, namely fatigue, memory or concentration impairment, achy muscles, absence of smell or taste senses, affective symptoms, and cognitive impairment, (c) symptoms should last for at least two months, and (d) symptoms should persist beyond the acute phase or become apparent 2–3 months later. The current study comprises (a) a case-control study comparing Long COVID patients with healthy controls, and (b) a retrospective inception and single cohort study which included inflammatory measures during the acute phase of Long COVID some months earlier.

In the recruited individuals (all staff members) with Long COVID, the diagnosis of acute COVID-19 infection was made by specialists in the fields of clinical pathology and virology based on the following criteria: (a) positive IgM antibody reactivity against SARS-CoV-2, (b) positive test results of reverse transcription real-time polymerase chain reaction (rRT-PCR), (c) clinical signs of severe infection, such as loss of the senses of smell and taste, shortness of breath, coughing and fever, and (d) being quarantined, hospitalized and treated for COVID-19 infection in one of the official Iraqi COVID centers including Middle Euphrates Center for Cancer; Al-Sader Medical City of Najaf; Al-Najaf Teaching Hospital, Al-Hakeem General Hospital, Hasan Halos Al-Hatmy Hospital for Transmitted Diseases, and Imam Sajjad Hospital. During Long COVID, the subjects showed a negative PCR test and were free of any symptoms of acute COVID-19, such as dry cough, sore throat, shortness of breath, fever, night sweats, or chills.

The controls were selected from the same group of staff members employees as the Long COVID participants and matched to the latter in terms of age, gender and BMI. A Hamilton Depression Rating Scale (HAMD) [34] score of <9 was the criterion for participation in the control group, which included about 33% of individuals who reported minor mental symptoms such as low mood and anxiety as a consequence of their social isolation and lack of social ties. Controls were only included if they had a negative rRT-PCR test result and had never shown clinical symptoms of COVID-19 infection.

Long COVID and control individuals with a life-time history of major depressive disorder, bipolar disorder, generalized anxiety disorder, dysthymia, and panic disorder as well as schizophrenia, psycho-organic syndrome, and substance use disorders (except tobacco use disorder or TUD) were not included in the study. We also excluded Long COVID and control individuals with a lifetime history of neurodegenerative and neuroinflammatory diseases such as Parkinson’s or Alzheimer’s disease, chronic fatigue syndrome, multiple sclerosis, stroke or systemic (auto-)immune diseases such as type 1 diabetes mellitus, psoriasis, inflammatory bowel disease, rheumatoid arthritis, systemic lupus erythematosus. Subjects with hepatic or renal disease, pregnant women, and lactating people were also excluded from the study.

### 2.2. Clinical Assessments

A senior psychiatrist conducted a semi-structured interview 3–4 months after recovery from acute COVID-19 to obtain the socio-demographic and clinical characteristics of all patients and the same psychiatrist also assessed the controls using the same interview and rating scales. On the same day as the semi-structured interview, all subjects completed the WHO-QoL-BREF (World Health Organization Quality of Life Instrument-Abridged Version) [35]. This scale rates 26 items across four categories of HR-QoL: (1) Domain 1 or physical health: energy, sleep, fatigue, pain, discomfort, work capacity, activities of daily living, medication dependence, and mobility; (2) Domain 2 or psychological health: self-esteem, body image, learning, thinking, concentration, memory, negative and positive feelings, and beliefs (spirituality-religion-personal); (3) Domain 3 or social relationships: social support, sexual activity, and personal relationships; and (4) Domain 4 or environment: physical safety and security. We calculated the raw scores for the four domains using the WHO-QoL-BREF criteria. The psychiatrist evaluated the severity of several symptom domains including depression severity using the HAMD [34] and the Beck Depression Inventory-II (BDI-II) [36], chronic fatigue and fibromyalgia using the Fibro-fatigue (FF) scale [37], and anxiety severity using the Hamilton Anxiety Rating Scale (HAMA, Monheim, Germany) [38]. Furthermore, we used the rating scale items to create subdomain severity scores of the major symptoms. We divided the HAMD into two subdomains: pure depressive symptoms (pure HAMD), which is the sum of sad mood, feelings of guilt, suicidal thoughts, and loss of interest, and physiosom HAMD (physiosom HAMD), which is the sum of somatic anxiety, gastrointestinal (GIS) anxiety, genitourinary anxiety, and hypochondriasis. Similarly, two subdomains of the HAMA were computed: pure anxiety symptoms (pure HAMA), which were defined as the sum of anxious mood, tension, fears, anxiety, and anxious behavior during the interview, and physiosomatic HAMA symptoms (physiosom HAMA), which were defined as the sum of somatic sensory, cardiovascular, GIS, genitourinary, and autonomic symptoms. Furthermore, after removing items from the FF scale that represented cognitive and affective symptoms, a pure physiosomatic FF (pure FF) score was calculated as the sum of muscular pain, muscle tension, fatigue, autonomous symptoms, gastro-intestinal symptoms, headache, and a flu-like malaise. We also computed the sum of all pure depressive BDI-II (pure BDI) symptoms, excluding physiosomatic symptoms, such as sadness, discouragement about the future, feeling a failure, dissatisfaction, feeling guilty, feeling punished, being disappointed in oneself, being critical of oneself, suicidal ideation, crying, loss of interest, difficulty making decisions, and work inhibition. In previous studies [7,8,12,13], the physio-affective phenome of Long COVID was defined as the first factor extracted from pure FF and BDI, as well as pure and physiosom HAMA and HAMD scores. We also recorded the vaccinations that the subjects received (AstraZeneca, Pfizer, and Sinopharm) as well as the treatments during the acute phase, namely dexamethasone, ceftriaxone (antibiotic of the cephalosporin third generation), azithromycin (antibiotic), enoxaparin sodium (anticoagulant) and bromhexine (mucolytic drug). TUD was diagnosed using the Diagnostic and Statistical Manual of mental Disorders, 5th edition. Body mass index (BMI) was calculated by dividing weight in kilograms by height in meters squared.

### 2.3. Biomarker Assessments

The biomarkers of the acute phase of infection were peak body temperature (PBT) and the lowest SpO2 values. Patients’ records were used to acquire PBT and the lowest SpO2 values that were recorded while they were quarantined or hospitalized for the acute infection. Both the electronic oximeter (Shenzhen Jumper Medical Equipment Co. Ltd., Shenzhen, China) and a sublingual digital thermometer with beep sound were used by a qualified paramedical professional. A composite score based on lowered SpO2 but increased PBT was calculated by subtracting the z transformed SpO2 (z SpO2) values from z PBT values (dubbed as the TO2 index).

During Long COVID, 3–4 months after the acute phase, five milliliters of venous blood were taken in the early morning hours of 7.30–9.00 a.m., and immediately injected into serum tubes after the subjects had fasted overnight. We did not use any blood that had been hemolyzed, lipemic, or icteric. After 10 min of incubation at room temperature, all tubes were centrifuged at 3000× *g* rpm. For biochemical testing, we prepared three aliquots of serum and kept them at −70 °C in Eppendorf tubes. Agappe, Diagnostics Ltd. (Cham, Switzerland), provided ready-to-use kits for the spectrophotometric measurement of total serum calcium. Serum levels of IL-1β, IL-18, IL-10, caspase-1, MPO and advanced oxidation protein products (AOPP) (a marker of protein oxidation) were measured using kits from Nanjing Pars Biochem (Nanjing, China). Glucose was measured spectrophotometrically by a kit supplied by Biolabo^®^ (Maizy, France) and insulin using a commercial ELISA sandwich kit from DRG^®^ International Inc. (Springfield, NJ, USA). All inter-assay coefficients of variations of all analytes were <10%. Using the biomarker concentrations, we calculated z unit-based weighted composite scores including oxidative stress as z MPO + z AOPP (dubbed oxidative stress toxicity or OSTOX), NLRP3 inflammasome as z IL-1β + z IL-18 + z caspase 1 (dubbed NLRP3), and OSTOX+NLRP3 as zIL-1β + zIL-18 + z caspase 1 + z MPO + z AOPP. Insulin resistance was computed as z insulin + z glucose (dubbed zIR) and added to OSTOX+NLRP3 was denoted as neurotoxicity (NT). Finally, we also combined the biomarkers of acute inflammation (PBT and SpO2) together with the NT biomarkers into one composite (dubbed NT+TO2) computed as z PBT − zSpO2 + zIL-1β + zIL-18 + z caspase 1 + z MPO + z AOPP + zIR.

### 2.4. Statistics

Analysis of variance (ANOVA) and contingency tables were used to explore differences in scale or nominal variables between the research groups. The association between scale variables was checked using Pearson product-moment coefficients. This study employed univariate general linear models (GLMs) to assess how clinical rating scale scores and COVID biomarkers are associated with newly formed categories based on WHO-Qol data while accounting for variables such as age, gender, sex and BMI. Fisher’s protected Least Significant Difference (LSD) test was used to examine multiple groups mean differences (LSD) (protected means that the omnibus test should be significant prior to using the LSD test). K-means and two-step cluster analysis were applied to the 4 WHO-QoL domains to delineate novel clusters of subjects based on their WHO-QoL scores. The accuracy of the cluster solution was evaluated employing the silhouette cohesion and separation measure (>0.5 is deemed adequate). Multiple regression analysis was used to examine the potential of clinical rating scales and biomarkers of acute and Long COVID to predict WHO-QoL scores, while allowing for the effects of age, sex, education, BMI, treatments and vaccinations. We also used a forward stepwise automatic regression method with 0.05 and 0.06 *p*-values for inclusion and omission in the final regression model. We calculated the standardized beta-coefficients and t-statistics (with exact *p*-value) for each of the explanatory variables in the final recession models, and we also calculated F statistics (and *p* values) and total variance (R^2^ or partial eta squared, also used as effect size) explained by the model. Collinearity and multicollinearity were also evaluated using the variance inflation factor (cut off >4), tolerance (<0.25) and the condition index and variance proportions in the collinearity diagnostics table. Where needed, we grouped predictors in composites to solve collinearity problems or to reduce the number of features, for example, using the TO2 and NT indices. The White and modified Breusch-Pagan tests were used to verify the heteroskedasticity. All statistical analyses were conducted using IBM SPSS version 28 (IBM, Chicago, IL, USA). All tests are two-tailed, with a significance level of *p* =0.05.

To investigate the causative relationships between the inflammatory response of acute SARS-CoV-2 infection (the TO2 index), the biomarkers of Long COVID, the physio-affective phenome of Long COVID and HR-QOL, a method known as partial least squares (PLS) path analysis was utilized [24]. Our model assumes that the effects of the input variables (infection and TO2 index) on HR-QoL are partially mediated by the path from biomarkers of Long COVID (NT and calcium) to the physio-affective phenome. In addition, we added sex, age, BMI, and vaccination as additional input variables. All the variables that were used as input (e.g., sex, vaccination, TO2 index) and the long COVID biomarkers were entered as single indicators, whilst the physio-affective phenome and HR-QoL domain scores were entered as latent vectors. We derived a first latent vector from the values of pure and physiosom HAMA and HAMD, as well as pure FF and BDI (dubbed the physio-affective phenome), and a second latent vector from the 4 WHO-QoL raw scores (dubbed the HR-QoL latent vector). Only when the inner and inner models satisfied the following prespecified quality criteria was complete PLS path analysis carried out: (a) the output latent vectors demonstrate high construct and convergence validity as indicated by average variance extracted (AVE) > 0.5, rho A > 0.8, Cronbach’s alpha > 0.7, and composite reliability > 0.8, (b) all loadings on both extracted latent vectors are >0.6 at *p* < 0.001, (c) the overall model fit namely the standardized root square residual (SRMR) value is <0.08, (d) Confirmatory Tetrad Analysis (CTA) demonstrates that both latent vectors are not mis-specified as reflective models, (e) blindfolding demonstrates that the construct’s cross-validated redundancies are adequate, and (f) the model’s prediction performance is satisfactory as measured by PLS Predict. If all the above-mentioned model quality data meets the predetermined criteria, we run a complete PLS path analysis with 5000 bootstrap samples, produce the path coefficients (with exact *p*-values) and in addition, compute the specific indirect and total indirect (that is mediated) effects as well as the total effects.

### 2.5. Avoiding Bias

The retrospective identification of exposure biomarkers (SpO2 and PBT) was performed by chart reviewers who assessed patient records and were blinded from the outcome data. The target study population (Long COVID) was well defined as described above and we selected individuals who showed clinical signs of Long COVID coupled with a negative rRT-PCR and had suffered from confirmed (by rRT-PCR and symptoms) acute COVID-19 infection some months earlier. Interviewer bias was minimized because the senior psychiatrist interviewer was blinded from the exposure data (medical records) and the outcome (medical diagnoses of Long COVID and HR-QoL data). Bias from misclassification is excluded because exposure (acute infection) and outcome diagnosis (Long COVID) are based on laboratory and well-defined clinical data. Statistical analyses were controlled for diverse confounders including age, sex, education, and tobacco use. As reported, there were no conflicts of interest, and the study was not funded. Figure 1 shows the flow of the subjects from recruitment to statistical analysis. Electronic Appendix A shows the baseline characteristics of patients and controls indicating that there were no significant differences in age, sex, body mass index, education, marital status, residency, tobacco use disorder and vaccination status between the groups.

## 3. Results

### 3.1. Results of Cluster Analysis

In order to retrieve clusters of participants based on the four WHO-QoL domain scores, we performed cluster analysis whereby K-means showed the best solution with three clusters: a first cluster (*n* = 42) with normal WHO-QoL domain scores, a second cluster (*n* = 37) with moderately reduced WHO-QoL values, and a third cluster with very low WHO-QoL domain scores. The silhouette measure of cohesion and separation was good with a value of 0.6. Table 1 shows the scores on the four WHO-QoL domains as well as the first latent vector extracted from the four domains after covarying for age, sex, education, and treatments. There was a strong association between WHO-QoL groups and the diagnosis of acute COVID infection. There were no significant differences in age, sex, BMI, marital status, education, rural/urban ratio, TUD and vaccination among the three clusters. This table also shows the treatments during the acute phase of illness.

### 3.2. The Physio-Affective Phenome Scores and Biomarkers in WHO-QoL Clusters

Table 2 shows the physio-affective phenome scores and biomarkers in the WHO-QoL clusters. We found that all rating scale scores, either total scores or the subdomains (except physiosom HAMA) were significantly different between the three clusters and increased from the normal QoL to the moderately low QoL to the very low QoL cluster. The physiosom HAMA score was significantly higher in both the moderate and very low clusters as compared with the normal QoL cluster. PBT, TO2 index, OSTOX+NLRP3, and NT+TO2 were significantly different between the three clusters and increased from the normal WHO-QoL to the moderately low WHO-QoL to the very low WHO-QoL cluster. SpO2 values were significantly lower in those with moderate and very low WHO-QoL scores as compared with those with normal WHO-QoL, whilst OSTOX, NLRP3, and NT were significantly increased in the moderate and very low WHO-QoL groups.

### 3.3. Associations of the Physio-Affective Phenome with WHO-QoL Scores

Table 3 shows the results of multiple regression analyses with the WHO-QoL scores as dependent variables and the depression, anxiety, and FF scales as explanatory variables. Model#1 shows that 76.7% of the variance in the first latent vector extracted from the four subdomains was explained by the regression on pure BDI and FF and total HAMD scores (all inversely associated). Model#2 shows that 75.0% of the variance in the physical domain was explained by 4 predictors, namely pure FF and BDI, total HAMD and male sex. Up to 68.8% of the variance in the psychological domain (Model#3) was explained by pure BDI and FF (inversely associated). A smaller part of the variance (16.7%) of the social component was explained by the total HAMD score only, whilst 58.4% of the variance in the environmental component (Model#5) was explained by pure BDI and FF scores. Figure 2 shows the partial regression of the social subdomain of the WHO-QoL on the total BDI-II score. 

### 3.4. Associations of the Biomarkers with WHO-QoL Scores

Table 4 shows the results of multiple regression analyses with the WHO-QoL scores as dependent variables and the biomarkers as explanatory variables. Model#1 shows that 59.0% of the variance in the overall WHO-QoL score (first PC extracted from the four domains) was explained by PBT and the NT+TO2 index (both inversely) and calcium (positively). The physical subdomain was explained (57.9% in Model#2) by PBT and NT (inversely) and calcium (positively). Figure 3 shows the partial regression of the physical domain score on PBT. Model#3 shows that 39.9% of the variance in the psychological domain was predicted by NT+TO2 (inversely) and calcium (positively). Figure 4 shows the partial regression of the psychological domain score on the NT+TO2 index. Up to 26.1% of social WHO-QoL score was explained by the cumulative effects of calcium (positively) and NT (inversely). Model#5 shows that 48.6% of the variance in the environmental WHO-QoL domain was predicted by PBT and NT+TO2 (inversely) and calcium (positively). The same table also shows that the physio-affective phenome score was significantly associated with female sex, PBT and NT (positively associated) and calcium (inversely associated).

Table 5 lists some regression analyses which assessed the effects of being infected with the SARS-CoV-2 virus, the treatments during the acute phase and the physio-affective domains of Long COVID on the WHO-QoL domains. We found that 84.8% (Model#1) of the variance in the overall WHO-QoL score was predicted by infection, pure BDI and FF (inversely) and treatment with enoxaparin (positively). Model#2 shows that after removal of the affective phenome features, 68.6% of the variance in the physical WHO-QoL domain score was explained by infection, PBT, ceftriaxone treatment and the Astra vaccination (all inversely associated). Model#3 shows that 59.6% of the variance in the environmental component was explained by infection (inversely) and treatment with enoxaparin (positively).

Figure 5 shows the final PLS model which displays only the significant paths and indicators. With an SRMR of 0.047, the model quality is adequate. We observed adequate convergence and construct reliability validity values for (a) the HR-QoL latent factor with AVE = 0.721, composite reliability = 0.910, Cronbach alpha = 0.867, and rho_A = 0.910, while all loadings were >0.669 at *p* < 0.001); and (b) the physio-affective phenome, with AVE = 0.610, composite reliability = 0.904, Cronbach alpha = 0.873, rho_A = 0.887, while the loadings of the physio-affective phenome variables were >0.704. Blindfolding revealed acceptable construct redundancies of 0.491 for the HR-QoL latent vector and 0.365 for the physio-affective phenome latent vector. According to CTA, both latent vectors were correctly specified as reflective models. According to PLSPredict, all construct indicators had positive Q^2^ predict values, indicating that the prediction error was less than the naivest benchmark. PLS analysis with 5000 bootstraps showed that 70.3% of the variance in HR-QoL was explained by the regression on the physio-affective phenome, calcium and NT, whilst 61.5% of the variance in the physio-affective phenome was explained by calcium, NT, TO2, vaccination (Astra-Zeneca and Pfizer) and female sex. Moreover, TO2 significantly affected calcium (inversely) and NT (positively). All five specific indirect effects of TO2 (and thus also SARS-CoV-2 infection) on HR-QoL were significant yielding highly significant total indirect (t = −5.03, *p* < 0.001) and total (t = −13.30, *p* < 0.001) effects. The effects of sex (t = 2.66, *p* = 0.004) and vaccination (t = −2.35, *p* = 0.009) on HR-QoL were mediated via effects on the physio-affective phenome. Finally, the NT index yielded not only direct effects but also specific indirect effects (t = −3.16, *p* < 0.001) and thus a highly significant total effect (t = −3.86, *p* < 0.001).

We have also examined if we could combine the six phenome domains and the four WHO-QoL domains into one latent vector. We found that indeed one replicable vector could be extracted from the ten indicators with AVE = 0.593, rho_A = 0.931, with sufficient loadings (all >0.6 at *p* < 0.001 except for the social domain which showed a loading of 0.552). Blindfolding revealed an acceptable construct redundancy of 0.379. Complete PLS analysis performed using 5000 bootstrap samples showed that 65.5% of the variance in this physio-affective-HR-QoL factor was explained by female sex (*p* = 0.03), vaccination (*p* = 0.012), TO2 (*p* < 0.001), calcium (*p* < 0.001) and the neurotoxicity index (*p* < 0.001).

## 4. Discussion

### 4.1. Lowered HR-Qol in Long COVID

The first major finding of this study is that individuals with Long COVID had significantly lower total HR-QoL scores as well as physical, psychological, and environmental QoL scores, but not social QoL scores. These findings support recent systematic review and meta-analysis studies that show HR-QoL is significantly lower in Long COVID-19 and that 18 to 60% of people with Long COVID have decreased HR-QoL [28,29,30]. In this context, we discovered that approximately 55% of people with Long COVID have extremely low physical, psychological, and environmental QoL scores. Comparing our findings to those obtained in major depressive disorder and bipolar disorder reveals a strong parallel, except for the social QoL domain, which was significantly decreased in 50% of MDD/BD patients, whereas Long COVID patients did not show a significant decrease in social QoL [39]. Thus, 50% of MDD/BD patients have very low levels of the physical domain (mean ± SD: 19.3 ± 4.3) compared to 55% of Long COVID patients who show a mean (SEM) score of 16.8 (0.49). In contrast, the psychological domain was lower in the MDD/BD patients with the lowest mean ±SD scores, namely (14.7 ± 3.1), versus Long COVID (mean, SEM: 16.4 ± 0.42). Such disparities may be attributed to the greater impact of SARS-CoV-2 infection on the physical domain, and affective disorders on the psychological domain. Both MDD/BD and Long COVID patients had similar environmental QoL domain scores [39].

### 4.2. The Physio-Affective Phenome of Long COVID Predict Lowered HR-QoL

The second major finding of this study is that in Long COVID, there were strong associations between lower HR-QoL scores and higher physio-affective domain scores, which included depression, fatigue, and physio-somatic symptoms, as well as anxiety. In fact, the cumulative effects of the pure BDI, pure FF, and total HAMD score explained a large portion of the variance in total QoL (around 76.7 percent) and domain scores (16.7–75 percent). As a result, a combination of pure affective, pure physiosomatic symptoms, and chronic fatigue determines HR-QoL in Long COVID to a large extent. These findings support previous research that Long COVID patients with affective (such as sadness and anxiety) and physiosomatic (such as fatigue, myalgia, and joint pain) symptoms have significantly lower HR-QoL scores on physical and mental health [26]. In patients suffering from MDD/BD we previously discovered that the total WHO-QoL score, as well as the four subdomain scores, were strongly and inversely related to the total HAMD and HAMA scores [39,40], with the HAMD having a much stronger effect on the total WHO-QoL scores than the HAMA [40]. In a Thai schizophrenia patient population, we discovered that total FF and HAMA scores explained a large portion of the variance in the total WHO-QoL score, while the four subdomain scores were inversely related to the HAMA (physical, social, and environmental) and the FF (psychological and environmental) scores [41]. Overall, both affective and physiosomatic symptoms, including chronic fatigue, contribute significantly to lower HR-QoL in Long-COVID.

### 4.3. Lowered HR-Qol in Long COVID Is Predicted by Neuroimmunotoxic and Oxidative Pathways 

The third major finding of this study is that biomarkers of acute and Long COVID strongly predict lower WHO-QoL scores in Long COVID. Thus, increased PBT and the TO2 index, reflecting the immune-inflammatory response during the acute infectious phase (see Introduction), and Long-COVID biomarkers, namely increased neurotoxicity (due to increased NLRP3, OSTOX, and IR combined), and lowered calcium, strongly predict all WHO-QoL domain scores. In patients with MDD/BD, lower WHO-QoL scores were strongly associated with neuro-oxidative toxicity markers such as peroxides, malondialdehyde, superoxide dismutase, nitric oxide, and AOPPs, as well as lower HDL-cholesterol and paraoxonase 1, an antioxidant enzyme [39]. Kanchanatawan et al. [41] discovered that the total WHO-QoL score was associated with indices of tryptophan catabolite (TRYCAT) pathway activation with increased production of neurotoxic TRYCATs such as picolinic acid, xanthurenic acid, and 3-OH-kynurenine in Thai schizophrenia patients. Al-Musawi et al. [42] discovered a significant inverse relationship in Iraqi schizophrenia patients between total WHO-QoL scores and the pathogenic Thelper-17 (Th-17) phenotype and the IL-6/IL-23/Th-17 axis, which has major neurotoxic effects. Furthermore, increased levels of IL-1β, IL-6, IL-17, IL-21, IL-22, IL-23, and tumor necrosis factor (TNF)-α were all found to be inversely related to HR-QoL in these schizophrenia patients. Overall, it appears that decreased HR-QoL in Long COVID and other neuro-immune disorders is caused, at least in part, by increased toxicity due to neuro-immune and neuro-oxidative stress pathways.

### 4.4. The Effects of SARS-CoV-2 Infection on Lowered HR-QoL in Long COVID Are Mediated by Acute and Chronic Immune-Inflammatory Processes

The fourth major finding of this study is that the effects of SARS-CoV-2 infection and the severity of the immune-inflammatory response during the acute phase on HR-QoL were significantly mediated by increased neurotoxicity and decreased calcium (both of which were determined at least in part by the acute inflammation) and the effects of those biomarkers on the physio-affective phenome, which in turn affects HR-QoL. As such, this study discovered multiple mediated causal paths from SARS-CoV-2 infection to activation of immune-inflammatory pathways, increased neurotoxicity, and decreased calcium to the phenome, and, as a result, HR-QoL. We previously developed comparable multistep mediated models to explain decreased HR-QoL in schizophrenia. Al-Musawi et al. [42], for example, discovered that the neurotoxic effects of the IL-6/IL-23/Th-17 axis on the four WHO-QoL domain scores combined with disability scores were significantly and partially mediated by effects of the neurotoxic axis on the symptomatome of schizophrenia. In another study in schizophrenia, we discovered that the effects of neurotoxic TRYCATs on WHO-QoL scores were mediated by the schizophrenia phenome, as measured by key symptoms of schizophrenia and neurocognitive deficits [43].

Furthermore, in the current study, we were able to extract one latent construct from the four WHO-QoL subdomains and the six domains of the physio-affective core, combining the physio-affective symptomatome and phenomenome (or HR-QoL) data into a single phenome core. Notably, the TO2 index, NT index, and lower calcium explained 65.5% of the variance in this combined phenome core. Maes et al. [44] combined HAMD and HAMA scores, as well as the four WHO-QoL domains and Sheehan disability scores, in MDD/BD patients to create one latent construct that was highly predicted by OSTOX neurotoxicity and decreased antioxidant activity of the HDL-paraoxonase 1 complex [44].

We have previously discussed how various neuroimmunotoxic and neuro-oxidative pathways may contribute to the physio-affective phenome of Long COVID [7,8,12,13], MDD and BD [44,45], schizophrenia [31,33], and ME/CFS [46]. According to the theory summarized in this work, activated neuro-immune and neuro-oxidative pathways cause dysfunctions in peripheral and central neuronal cells as well as central circuits that mediate affection, sleep, pain, cognition, and memory. Recently, we discovered that in MDD, the effects of peripheral inflammation (as measured by CRP), lower calcium, and insulin resistance on the physio-affective phenome of depression (as measured by HAMD, HAMA, and FF scores) were mediated by neuronal injury indicators indicating damage to astroglial and neuronal (axonal) projections [47]. In patients with unstable angina, we discovered that activation of immune-inflammatory pathways (CRP and cytokines including IL-6) affects the physio-affective phenome of unstable angina, and that these effects are mediated by increased atherogenicity and insulin resistance [48]. Insulin resistance and lowered calcium in conjunction with immune biomarkers, also predict the physio-affective phenome in type 2 diabetes [49]. Previously, we discussed the neurotoxic effects of increased insulin resistance, including increased permeability of the blood-brain barrier, decreased levels of brain-derived neurotrophic factor, impaired synaptic plasticity and dendritic spine damage, and decreased hippocampal volume and metabolic activity in the prefrontal cortex [47]. Lower serum calcium is not only an indicator of an inflammatory response [47], but it is also associated with physiosomatic symptoms including muscle spasms and cramps, neuromuscular irritability, paresthesia, circumoral numbness, neurocognitive and memory impairments, fatigue, and depression and anxiety [50,51,52]. Recent meta-analysis findings indicate that low calcium in COVID-19 patients is associated with increased severity, higher mortality, and more complications [53].

### 4.5. Additional Explanatory Variables

This study also discovered that female sex and vaccination with AstraZeneca and Pfizer were linked to increased severity of the Long-COVID physio-affective phenome. Similar sex effects were observed in another study on Long COVID [13]. Women have significantly higher rates of anxiety, depression, and chronic fatigue than men [54,55,56]. Men have more severe acute respiratory syndrome and critical COVID-19 during acute COVID-19 infection, whereas women have more sickness symptoms and fatigue [20]. Such differences may be explained by the effects of different neuro-immune pathways, with males having a more activated NLRP3 inflammasome, which leads to severe acute respiratory syndrome [20], and females having increased cytokine-induced activation of indoleamine-2,3-dioxygenase, which is associated with acute COVID-19, depression, anxiety, and chronic fatigue [57,58].

Previously, we discovered [7,8] that vaccination with Pfizer (mRNA-based) and AstraZeneca (viral vector with genetically engineered virus), but not with the Sinopharm (inactivated virus-based) vaccine may exacerbate the physio-affective phenome of Long COVID. In a few previous studies, SARS-CoV-2 vaccinations were linked to Long COVID-like symptoms such as anxiety, fatigue, sadness, and deficits in type 1 interferon signaling, autoimmune responses, increased synthesis of spike protein and T cell activation [59,60].

Furthermore, we found that treatment with enoxaparin during the acute phase may have a protective effect against the physio-affective phenome, whereas treatment with ceftriaxone is associated with a worsening of the phenome. While enoxaparin treatment has anticoagulant and antithrombotic properties, which may help to prevent blood clotting caused by SARS-CoV-2 infection, it also has anti-inflammatory and neuroprotective properties [61,62,63,64]. Treatment with ceftriaxone, a third-generation cephalosporin antibiotic, has been linked to gut dysbiosis [65], which has been linked to increases in the physio-affective phenome of MDD/BD, ME/CFS, and schizophrenia [24,33,46].

### 4.6. Limitations

This study would have been more interesting if we had also measured other pro-inflammatory cytokines of the M1 macrophage, Th-1, and Th-2 phenotypes and the IL-6/IL-23/Th-17 axis, growth factors and TRYCATs during Long COVID, in addition to additional assays of oxidative stress (e.g., xanthine oxidase, chlorinative stress biomarkers) and nitrosylyation. It could be argued that the relatively smaller sample size would render the parameter estimates of the regression analyses less precise. However, increasing the number of participants would entail a larger number of plates to assay the biomarkers and thus an increasing analytical error due to the increasing inter-assay (and inter-plate) variation which may significantly decrease the overall precision (especially when measuring cytokines at the lower concentration ranges) [66]. The present study was performed in an Iraqi population and, therefore, may not have sufficient generalizability to other populations or ethnicities. Therefore, our results deserve to be replicated in other countries and ethnicities. The strength of this study is that the precision nomothetic approach allowed to delineate the effects of inflammation during the acute phase of COVID-19 on the phenome and lowered HR-QoL in Long COVID, and that these effects are mediated by the NLRP3 and oxidative stress pathways.

## 5. Conclusions

The severity of the immune-inflammatory response during the acute infection phase, which generates greater neuroimmunotoxicity and neuro-oxidative toxicity, predicts the physio-affective phenome and, as a result, reduced HR-QoL in Long COVID. During the acute phase, treatment with enoxaparin, which has antithrombotic, anti-inflammatory, and neuroprotective effects, was linked with an improvement in HR-QoL. These results suggest that interventions throughout the acute and long COVID phases that improve neuroprotection and inflammation, and target neurotoxicity may be clinically effective in preventing Long COVID physio-affective symptoms.

## Figures and Tables

**Figure 1 ijerph-19-10362-f001:**
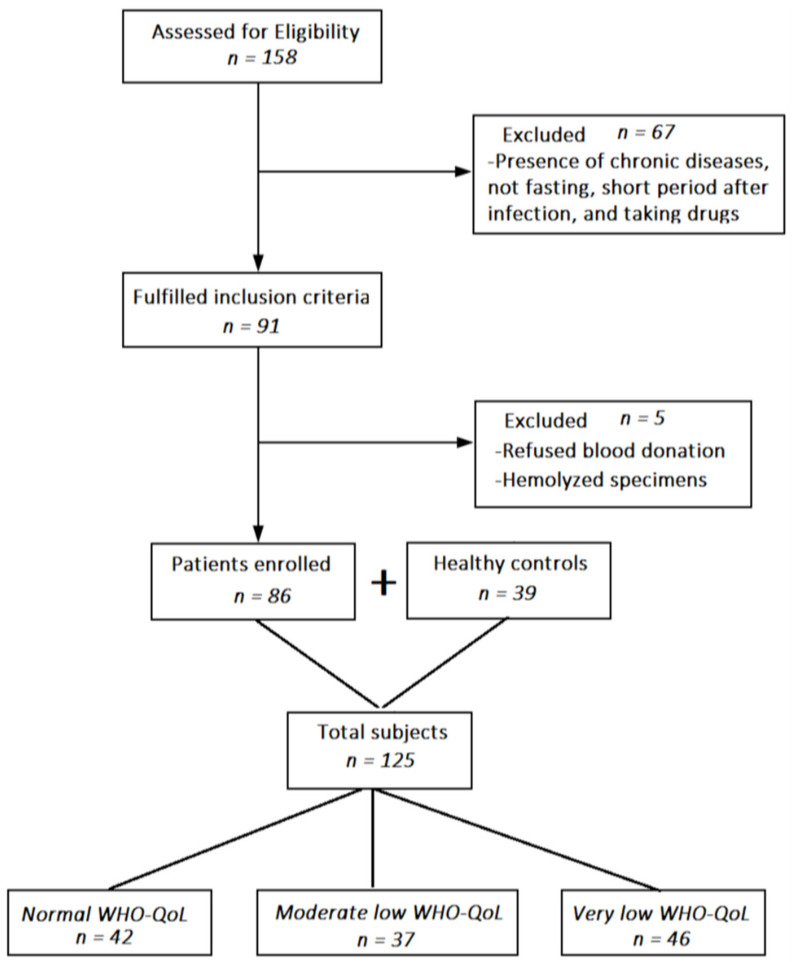
Flow chart depicting the flow of the participants from recruitment to statistical analysis.

**Figure 2 ijerph-19-10362-f002:**
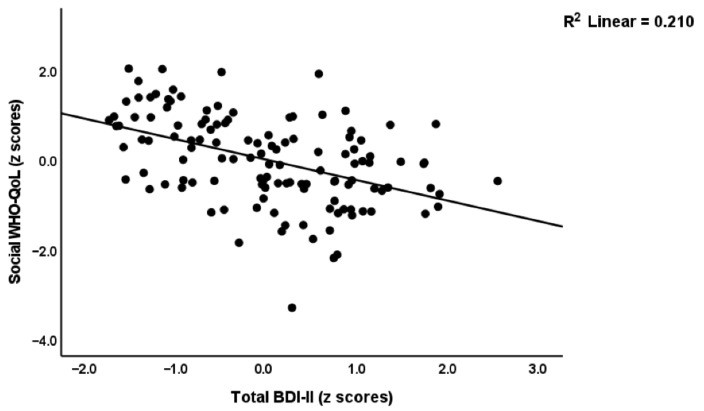
Partial regression of the social subdomain of the World Health Organization Quality of Life I(WHO-QoL) Instrument-Abridged Version score on the total Beck Depression Inventory (BDI-II) score.

**Figure 3 ijerph-19-10362-f003:**
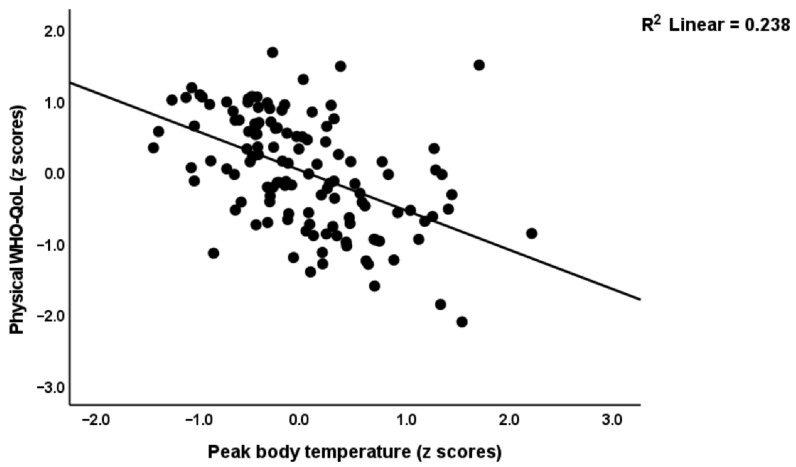
Partial regression of the physical subdomain of the World Health Organization Quality of Life I(WHO-QoL) Instrument-Abridged Version score on peak body temperature during the acute phase of illness.

**Figure 4 ijerph-19-10362-f004:**
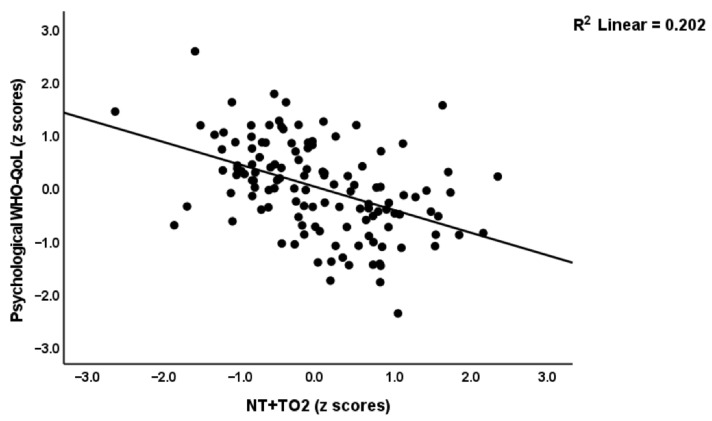
Partial regression of the psychological domain of the World Health Organization Quality of Life I(WHO-QoL) Instrument-Abridged Version score on an index of inflammation (TO2) during acute infection and neurotoxicity (NT) during Long COVID (NT+TO2).

**Figure 5 ijerph-19-10362-f005:**
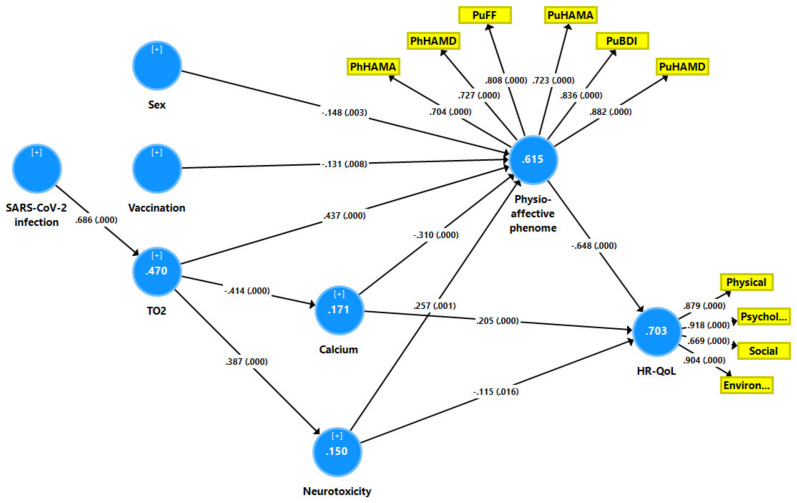
Results of partial least squares (PLS) analysis. Health related quality of life (HR-QoL) is entered as a latent vector extracted from 4 QoL domains, namely physical, psychological, social and environmental. The physio-affective phenome of Long COVID is entered as a latent vector extracted from 6 clinical domains, namely the pure Fibro-Fatigue (PuFF), Hamilton Depression (PuHAMD) and Anxiety (PuHAMA) rating scale scores, pure Beck Depression Inventory (PuBDI) scores, and physiosomatic HAMD (PhHAMD) and HAMA (PhHAMA) scores. All other variables were entered as single indicators, namely sex (men = 1 and women = 0), vaccination (Astra-Zeneca and Pfizer = 1 and Sinopharm = 0), calcium, neurotoxicity (NT, a combination of inflammation + insulin resistance + oxidative stress) and TO2 (index of increased peak body temperature and lower oxygen saturation). Shown are path coefficients (with *p*-values) and loadings (with *p*-values) on the latent vectors and variance explained (white figures in blue circles).

**Table 1 ijerph-19-10362-t001:** Socio-demographic and clinical variables in healthy controls (HC) and subjects with Long COVID divided into those with normal, moderately low and very low health-related quality of life (QoL) scores as measured with the Health Organization Quality of Life Instrument-Abridged Version (WHO-QoL).

Parameter	Normal WHO-QoL ^A^ *n* = 42	Moderate Low WHO-QoL ^B^ *n* = 37	Very Low WHO-QoL ^C^ *n* = 46	F/χ^2^	df	*p*
WHO-QoL, physical *	27.46 ± 0.66 ^B,C^	21.60 ± 0.51 ^A,C^	16.83 ± 0.49 ^A,B^	66.83	2/114	<0.001
WHO-QoL, psychological *	25.70 ± 0.57 ^B,C^	21.40 ± 0.43 ^A,C^	16.43 ± 0.42 ^A,B^	77.50	2/114	<0.001
WHO-QoL, social *	11.92 ± 0.43	10.58 ± 0.33	10.56 ± 0.32	2.50	2/114	0.086
WHO-QoL, environment *	33.60 ± 0.71 ^B,C^	26.78 ± 0.54 ^A,C^	22.89 ± 0.53 ^A,B^	54.03	2/114	<0.001
PC 4 WHO-QoL domains *	1.121 ± 0.087	−0.124 ± 0.066	−0.924 ± 0.064	135.31	2/114	<0.001
HC/Long COVID	38/4	1/36	0/46	FFHE		<0.001
Age (years)	28.0 ± 7.4	29.3 ± 6.5	27.9 ± 5.9	0.35	2/127	0.706
Female/Male ratio	19/23	15/22	20/26	0.18	2	0.914
BMI (kg/m^2^)	25.84 ± 4.08	25.83 ± 3.53	26.21 ± 5.23	0.05	2/127	0.950
Education (years)	15.0 ±1.2 ^B,C^	15.8 ± 1.9 ^A,C^	15.6 ± 1.7 ^A,B^	9.99	2/127	<0.001
Married/Single (No/Yes)	19/23	21/31	15/21	0.23	2	0.901
Rural/Urban (No/Yes)	8/34	8/29	7/39	0.58	2	0.749
TUD (No/Yes)	29/13	24/13	32/14	2.40	2	0.887
Vaccination A/Pf/S	9/23/10	5/23/9	15/23/8	4.46	4	0.347
Dexamethasone (No/Yes)	39/3	24/13	23/23	19.17	2	<0.001
Ceftriaxone (No/Yes)	41/1	18/19	16/30	38.94	2	<0.001
Azithromycine (No/Yes)	38/4	17/20	25/21	19.87	2	<0.001
Enoxaparin sodium (No/Yes)	38/4	4/33	8/38	67.52	2	<0.001
Bromhexine (No/Yes)	39/3	10/27	8/38	57.71	2	<0.001

Data are shown as mean (SD) (except: * shown as estimated marginal means and SE after adjusting for confounders) or as ratios. F: results of analysis of variance; χ^2^: results of analysis of contingency tables; FFHE: Fisher-Freeman-Halton Exact test. ^A, B, C^: Pairwise comparison among group means. BMI: body mass index, TUD: tobacco use disorder, vaccination A/Pf/S: AstraZeneca, Pfizer and Sinopharm.

**Table 2 ijerph-19-10362-t002:** Neuropsychiatric rating scale scores and biomarkers in healthy controls (HC) and subjects with Long COVID divided into those with normal, moderately low and very low health-related quality of life (QoL) scores as measured with the Health Organization Quality of Life Instrument-Abridged Version (WHO-QoL).

Variables	Normal WHO-QoL ^A^ *n* = 42	Moderate Lower WHO-QoL ^B^ *n* = 52	Very Low WHO-QoL ^C^ *n* = 36	F (df = 2/122)	*p*
Total FF score	11.0 ± 4.1 ^B,C^	20.4 ± 10.1 ^A,C^	36.0 ± 12.1 ^A,B^	78.42	<0.001
Total HAMA score	7.9 ± 3.9 ^B,C^	13.8 ± 6.6 ^A,C^	19.7 ± 8.5 ^A,B^	34.26	<0.001
Total BDI-II score	9.1 ± 4.1 ^B,C^	20.3 ± 5.8 ^A,C^	28.9 ± 6.4 ^A,B^	140.46	<0.001
Total HAMD score	6.4 ± 3.7 ^B,C^	14.5 ± 4.8 ^A,C^	18.8 ± 4.5 ^A,B^	90.23	<0.001
Pure FF	−0.867 ± 0.385 ^B,C^	−0.079 ± 0.746 ^A,C^	0.855 ± 0.849 ^A,B^	68.31	<0.001
Pure HAMD	−0.987 ± 0.395 ^B,C^	0.136 ± 0.636 ^A,C^	0.792 ± 0.851 ^A,B^	80.21	<0.001
Physiosom HMD	−0.862 ± 0.672 ^B,C^	0.247 ± 0.949 ^A,C^	0.588 ± 0.726 ^A,B^	40.35	<0.001
Pure HAMA	−0.547 ± 0.766 ^B,C^	−0.084 ± 0.853 ^A,C^	0.568 ± 1.012 ^A,B^	17.53	<0.001
Physiosom HAMA	−0.517 ± 0.564 ^B,C^	0.002 ± 0.958 ^A^	0.470 ± 1.120 ^A^	12.73	<0.001
Pure BDI	−0.998 ± 0.605 ^B,C^	0.209 ± 0.663 ^A,C^	0.743 ± 0.735 ^A,B^	76.11	<0.001
PC Physio-affective phenome	−0.963 ± 0.368 ^B,C^	0.0498 ± 0.706 ^A,C^	0.839 ± 0.804 ^A,B^	82.90	<0.001
Peak body temperature	37.07 (0.78) ^B,C^	38.30 (0.74) ^A,C^	38.75 (0.93) ^A,B^	47.85	<0.001
Lowest SpO2 (%)	94.86 ± 1.96 ^B,C^	91.62 ± 3.59 ^A^	90.37 ± 4.29 ^A^	19.46	<0.001
TO2 index (zBT-zSpO2 in z scores)	−0.880 ± 0.586 ^B,C^	0.218 ± 0.759 ^A,C^	0.628 ± 0.903 ^A,B^	44.71	<0.001
NLRP3 (z scores)	−0.406 ± 0.945 ^B,C^	0.030 ± 0.833 ^A^	0.347 ± 1.052 ^A^	6.85	0.002
OSTOX (z scores)	−0.380 ± 1.018 ^B,C^	0.140 ± 0.478 ^A^	0.269 ± 1.057 ^A^	5.45	0.005
OSTOX+NLRP3 (z scores)	−0.527 ± 0.880 ^B,C^	0.088 ± 0.794 ^A,C^	0.492 ± 0.905 ^A,B^	15.23	<0.001
zIR (z scores)	−0.426 ± 0.678 ^B,C^	0.307 ± 1.112 ^A^	0.142 ± 1.040 ^A^	6.57	0.002
OSTOX+NLRP3+IR (NT)	−0.625 ± 0.892 ^B,C^	0.188 ± 0.741 ^A^	0.419 ± 1.010 ^A^	16.04	<0.001
NT+TO2 (z scores)	−0.857 ± 0.795 ^B,C^	0.240 ± 0.677 ^A,C^	0.589 ± 0.851 ^A,B^	39.76	<0.001

Data are shown as mean (SD) or as ratios. F: results of analysis of variance. ^A, B, C^: Pairwise comparison among group means. BDI: Beck Depression Inventory; FF: Fibro-Fatigue scale, HAMD: Hamilton Depression rating Scale; NLRP3: index comprising interleukin-1β, IL-18 and caspase-1, advanced oxidation protein products and myeloperoxidase and insulin resistance (IR), OSTOX: index reflecting oxidative toxicity, IR: insulin resistance index.

**Table 3 ijerph-19-10362-t003:** Results of multiple regression analyses with the health-related quality of life (QoL) scores as measured with the Health Organization Quality of Life Instrument-Abridged Version (WHO-QoL) domain scores as dependent variables and physio-affective scores as explanatory variables.

Dependent Variables	Explanatory Variables	B	t	*p*	F Model	df	*p*	R^2^
**PC_WHO-QoL domains**	**Model#1**				**132.74**	**2/121**	**<0.001**	**0.767**
	Pure BDI	−0.476	−6.96	<0.001				
	Pure FF	−0.292	−5.04	<0.001				
	Total HAMD	−0.238	−3.02	0.003				
**WHO-QoL physical**	**Model#2**				**90.00**	**3/120**	**<0.001**	**0.750**
	Pure FF	−0.523	−8.63	<0.001				
	Pure BDI	−0.254	−3.56	<0.001				
	Sex	−0.125	−2.72	0.007				
	Total HAMD	−0.221	−2.69	0.008				
**WHO-QoL psychological**	**Model#3**				**133.17**	**2/122**	**<0.001**	**0.686**
	Pure BDI	−0.619	−10.41	<0.001				
	Total FF	−0.316	−5.31	<0.001				
**WHO-QoL social**	**Model#4**				**24.61**	**1/123**	**<0.001**	**0.167**
	Total HAMD	−0.408	−4.96	<0.001				
**WHO-QoL environmental**	**Model#5**				**85.53**	**2/122**	**<0.001**	**0.584**
	Pure BDI	−0.586	−8.78	<0.001				
	Pure FF	−0.283	−4.24	<0.001				

BDI: Beck Depression Inventory; FF: Fibro-Fatigue scale, HAMD: Hamilton Depression Rating Scale.

**Table 4 ijerph-19-10362-t004:** Results of multiple regression analyses with the health-related quality of life (QoL) scores as measured with the Health Organization Quality of Life Instrument-Abridged Version (WHO-QoL) domain scores as dependent variables and biomarkers of acute and Long COVID as explanatory variables.

Dependent Variables	Explanatory Variables	B	t	*p*	F Model	df	*p*	R^2^
**PC_WHO-QoL domains**	**Model#1**				**57.47**	**3/120**	**<0.001**	**0.590**
	PBT	−0.290	−3.02	0.003				
	Calcium	0.302	4.57	<0.001				
	NT+TO2	−0.329	−3.56	<0.001				
**WHO-QoL physical**	**Model#2**				**54.93**	**3/120**	**<0.001**	**0.579**
	PBT	−0.465	−6.22	<0.001				
	Calcium	0.256	3.82	<0.001				
	NT	−0.241	−3.58	<0.001				
**WHO-QoL psychological**	**Model#3**				**40.13**	**2/121**	**<0.001**	**0.399**
	NT+TO2	−0.446	−5.83	<0.001				
	Calcium	0.305	3.99	<0.001				
**WHO-QoL social**	**Model#4**				**21.33**	**2/121**	**<0.001**	**0.261**
	Calcium	0.401	5.04	<0.001				
	NT	−0.251	−3.16	0.002				
**WHO-QoL environmental**	**Model#5**				**37.87**	**3/120**	**<0.001**	**0.486**
	PBT	−0.276	−2.57	0.011				
	Calcium	0.288	3.89	<0.001				
	NT+TO2	−0.274	−2.65	0.009				
**PC phenome**	**Model#6**				**47.71**	**4/119**	**<0.001**	**0.616**
	PBT	0.480	6.70	<0.001				
	Calcium	−0.266	−4.13	<0.001				
	Female sex	−0.206	−3.61	<0.001				
	NT	0.223	3.45	<0.001				

PC_WHO-QoL 4 domains: principal component extracted from the 4 domains of the WHO-QoL-BREF (World Health Organization Quality of Life Instrument-Abridged Version) scale, PBT: peak body temperature, NT: neurotoxicity index comprising interleukin-1β, IL-18 and caspase-1, advanced oxidation protein products and myeloperoxidase and insulin resistance (IR), TO2: index of increased PBT and lowered oxygen saturation (SpO2).

**Table 5 ijerph-19-10362-t005:** Results of multiple regression analyses with the health-related quality of life (QoL) scores as measured with the Health Organization Quality of Life Instrument-Abridged Version (WHO-QoL) domain scores as dependent variables and being infected with the SARS-CoV-2 virus, treatments, vaccinations and clinical data as explanatory variables.

Dependent Variables	Explanatory Variables	B	t	*p*	F Model	df	*p*	R^2^
**PC_WHO-QoL 4 domains**	**Model#1**				**167.94**	**4/120**	**<0.001**	**0.848**
	Acute infection	−0.644	−8.50	<0.001				
	Pure BDI	−0.354	−6.95	<0.001				
	Pure FF	−0.274	−6.10	<0.001				
	Enoxaparin	0.262	4.08	<0.001				
**WHO-QoL physical**	**Model#2**				**65.39**	**4/120**	**<0.001**	**0.686**
	Acute infection	−0.525	−6.49	<0.001				
	PBT	−0.238	−3.16	0.002				
	Ceftriaxone	−0.165	−2.68	0.008				
	Vaccination A	−0.103	−2.00	0.048				
**WHO-QoL environmental**	**Model#3**				**89.82**	**2/122**	**<0.001**	**0.596**
	Acute infection	−1.054	−10.35	<0.001				
	Enoxaparin	0.379	3.72	<0.001				

PC_WHO-QoL 4 domains: principal component extracted from the 4 domains of the WHO-QoL-BREF (World Health Organization Quality of Life Instrument-Abridged Version) scale, BDI: Beck Depression Inventory; FF: Fibro-Fatigue scale; PBT: peak body temperature 3.5. Results of PLS Analysis.

## Data Availability

The dataset generated during and/or analyzed during the current study will be available from the corresponding author (M.M.) upon reasonable request and once the dataset has been fully exploited by the authors.

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
