# Peer review of "Lowered Quality of Life in Long COVID Is Predicted by Affective Symptoms, Chronic Fatigue Syndrome, Inflammation and Neuroimmunotoxic Pathways"

_ijerph, 2022, doi:10.3390/ijerph191610362_

Round 1
Reviewer 1 Report
Minor revision-
The title is good, but it could be shorter.
In the methodology, it is mandatory that authors add in the first paragraph the approval document number from the ethics committee for human studies.
The relationship of inflammation to the effects of long covid was correlated, but the predictors used were few. Is there a possibility of performing more inflammatory markers in an ELisa panel, or even an RT-PCRq for different genes?
Author Response
Minor revision-
The title is good, but it could be shorter.
@ANSWER: title is reduced to 19 words
In the methodology, it is mandatory that authors add in the first paragraph the approval document number from the ethics committee for human studies.
@@ANSWER: done
The relationship of inflammation to the effects of long covid was correlated, but the predictors used were few. Is there a possibility of performing more inflammatory markers in an ELisa panel, or even an RT-PCRq for different genes?
@@ANSWER: the predictors were relatively few but nevertheless explain 50% of the variance in the outcome data. It will be difficult to improve this prediction. Based on our knowledge, we selected the predictors with the highest chance to be relevant (it is not a high-risk study !!!). In the limitations section we now state that: This study would have been more interesting if we had also measured other pro-inflammatory cytokines of the M1 macrophage, Th-1, and Th-2 phenotypes and the IL-6/IL-23/Th-17 axis, and growth factors …
Reviewer 2 Report
This is an important article that discusses the factors that affect HR-QoL in people with acute COVID-19 infection versus people suffering from the post-COVID-19 syndrome. From what I understood, the authors used a case-control study design (evidenced by the presence of cases and controls, and the hint about the study being a retrospective one). Although the authors have done great work in their presentation of their data, I have some considerations:
Major considerations:
1. Please mention the study design in the methods.
2. Line 127: The authors said that this study is a "hybrid". Please explain the meaning of this in the same/following sentence.
3. The authors have sufficiently explained everything about the recruitment process of the cases. However, they forgot to discuss the same thing for the controls. We need to know how the controls were selected, did they get tested for COVID-19, what the selection criteria were, ... etc. Another important thing is to mention whether matching, for example, was done. A flow chart of the selection process will also aid in the understanding of the study's selection process.
4. The authors also need to discuss potential biases that were accounted for during the conduction of this study. This should also be addressed in the methods. Please note that this is different from the discussion about the possible biases that may have slipped into the study. The latter subject should be discussed in the limitations of the discussion.
5. I find it weird that the authors have not compared the baseline characteristics of the cases and controls in Table 1. This is like a standard procedure. Please explain the reason why or provide the table requested. It is also important for assessing the differences between the groups (for selection bias). Therefore, the appropriate statistical tests should be used as well.
6. Please discuss more limitations. For example, all biases that were not accounted for should be mentioned, and their implications should be discussed. Another example is the generalizability of the study (I believe there are some concerns about it due to the small sample size, which ultimately leads to inaccuracies in the regression models). Furthermore, the strengths of the study should also be highlighted. We need to know what this study has added and how it will guide future research.
Minor considerations:
1. The title is so long. Titles ranging from 10-20 (best 16) are associated with more reads and citations. Please consider reducing the word count. The authors may consider not giving away the conclusion of the study in its title. We should keep this to the press, not in academia.
2. I believe the authors have used a citation manager to manage the references. This is good. However, I believe they used the NLM format, which is good enough but has wide variability. This means that some papers will be referenced with PMID and/or PMCID while others will lack them. Moreover, the DOI of medRxiv pre-prints is listed in the references list as URLs, not sole DOIs. Please edit this. Finally, if the authors are using EndNote, they can download the MDPI referencing style from here (this will not edit the problem with medRxiv articles): https://endnote.com/style_download/mdpi/
Author Response
Major considerations:
- Please mention the study design in the methods.
- Line 127: The authors said that this study is a "hybrid". Please explain the meaning of this in the same/following sentence.
@@ANSWER (1 and 2): furher explained as:
The current study comprises a) a case-control study comparing Long COVID patients with healthy controls, and b) a retrospective inception and single cohort study which included inflammatory measures during the acute phase of Long COVID some months earlier.
- The authors have sufficiently explained everything about the recruitment process of the cases. However, they forgot to discuss the same thing for the controls. We need to know how the controls were selected, did they get tested for COVID-19, what the selection criteria were, ... etc. Another important thing is to mention whether matching, for example, was done. A flow chart of the selection process will also aid in the understanding of the study's selection process.
@@ANSWER: Inclusion criteria for controls are described as:
The controls were selected from the same group of staff members employees as the Long COVID participants and matched to the latter in terms of age, gender and BMI .A Hamilton Depression Rating Scale (HAMD) [34] score of <9 was the criterion for participation in the control group, which included about 33% of individuals who reported minor mental symptoms such as low mood and anxiety as a consequence of their social isolation and lack of social ties. Controls were only included if they had a negative rRT-PCR test result and had never shown clinical symptoms of COVID-19 infection.
Moreover, other exclusion criteria were described for patients as well as controls. A flow chart is now shown in the ESF (see Figure 1).
- The authors also need to discuss potential biases that were accounted for during the conduction of this study. This should also be addressed in the methods. Please note that this is different from the discussion about the possible biases that may have slipped into the study. The latter subject should be discussed in the limitations of the discussion.
@@ANSWER: We gave added a subsectin in the methods: 2.5. Avoiding bias
The retrospective identification of exposure biomarkers (SpO2 and PBT) was performed by chart reviewers who assessed patient records and were blinded from the outcome data. The target study population (Long COVID) was well defined as described above and we selected individuals who showed clinical signs of Long COVID coupled with a negative rRT-PCR and had suffered from confirmed (by rRT-PCR and symptoms) acute COVID-19 infection some months earlier. Interviewer bias was minimized because the senior psychiatrist interviewer was blinded from the exposure data (medical records) and the outcome (medical diagnoses of Long COVID and HR-QoL data). Bias from misclassification is excluded because exposure (acute infection) and outcome diagnosis (Long COVID) are based on laboratory and well-defined clinical data. Statistical analyses were controlled for diverse confounders including age, sex, education, and tobacco use. As reported, there were no conflicts of interest, and the study was not funded.
- I find it weird that the authors have not compared the baseline characteristics of the cases and controls in Table 1. This is like a standard procedure. Please explain the reason why or provide the table requested. It is also important for assessing the differences between the groups (for selection bias). Therefore, the appropriate statistical tests should be used as well.
@@ANSWER:
ESF Table 1 now present the baseline soci-demographc data of both study groups.
- Please discuss more limitations. For example, all biases that were not accounted for should be mentioned, and their implications should be discussed. Another example is the generalizability of the study (I believe there are some concerns about it due to the small sample size, which ultimately leads to inaccuracies in the regression models). Furthermore, the strengths of the study should also be highlighted. We need to know what this study has added and how it will guide future research.
@@ANSWER:
This study would have been more interesting if we had also measured other pro-inflammatory cytokines of the M1 macrophage, Th-1, and Th-2 phenotypes and the IL-6/IL-23/Th-17 axis, growth factors and TRYCATs during Long COVID in addition to additional assays of oxidative stress (e.g. xanthine oxidase, chlorinative stress biomarkers) and nitrosylyation.
It could be argued that the relatively smaller sample size would render the parameter estimates of the regression analyses less precise. However, increasing the number of participants would entail a larger number of plates to assay the biomarkers and thus an increasing analytical error due to the increasing inter-assay and inter-plate variation which may significantly decrease the overall precision (especially when measuring cytokines at the lower concencentration ranges) [66]. The present study was performed in an Iraqi population and, therefore, may not have sufficient generalizability to other populations or ethnicities. Therefore, our results deserve to be replicated in other countries and ethnicities. The strength of this study is that the precision medicine approach allowed to delineate the effects of inflammation during the acute phase of COVID-19 on the phenome and lowered HR-QoL in Long COVID, and that these effects are mediated by the NLRP3 and oxidative stress pathways.
Minor considerations:
- The title is so long. Titles ranging from 10-20 (best 16) are associated with more reads and citations. Please consider reducing the word count. The authors may consider not giving away the conclusion of the study in its title. We should keep this to the press, not in academia.
@@ANSWER: the title is shortened to 19 words
- I believe the authors have used a citation manager to manage the references. This is good. However, I believe they used the NLM format, which is good enough but has wide variability. This means that some papers will be referenced with PMID and/or PMCID while others will lack them. Moreover, the DOI of medRxiv pre-prints is listed in the references list as URLs, not sole DOIs. Please edit this. Finally, if the authors are using EndNote, they can download the MDPI referencing style from here (this will not edit the problem with medRxiv articles): https://endnote.com/style_download/mdpi/
@@ANSWER:
We have adjusted the DOI of medRxiv pre-prints and checked the PMID and PMCID citations.

Round 2
Reviewer 2 Report
Thank you for making the recommended amendments. There are some issues that need to be resolved before publication:
1. The authors have supposedly added a flow chart (Figure 1). The figure was not contained in the main manuscript but in the online supplementary data. I believe it should be added to the main manuscript.
2. Starting from line 408, the authors examine the results presented in Figure 5. There is no Figure 5. What are the authors explaining here precisely?
Regardless, I sincerely thank the authors for their awesome work.
Author Response
- The authors have supposedly added a flow chart (Figure 1). The figure was not contained in the main manuscript but in the online supplementary data. I believe it should be added to the main manuscript. @@ANSWER: now added to the main text.
2. Starting from line 408, the authors examine the results presented in Figure 5. There is no Figure 5. What are the authors explaining here precisely? @@ANSWERS: all figures are included and renumbered from 1-5.